# Effects of High-Hydrostatic-Pressure Treatment on Polyphenols and Volatile Aromatic Compounds in Marselan Wine

**DOI:** 10.3390/foods13152468

**Published:** 2024-08-05

**Authors:** Zicheng Yi, Danqing Zhao, Tengwen Chang, Xiang Chen, Jianrong Kai, Yang Luo, Bangzhu Peng, Binkun Yang, Qian Ge

**Affiliations:** 1Institute of Agricultural Product Quality Standards and Testing Technology, Yinchuan 750002, China; zcy06242024@163.com (Z.Y.); ctw19991008@163.com (T.C.); bzpeng@mail.hzau.edu.cn (B.P.); 2College of Food Science and Technology, Huazhong Agricultural University, Wuhan 430070, China; zdq_6264@163.com (D.Z.); wuyccx0413@163.com (X.C.); kaijianrong6688@163.com (J.K.); bkyang98@163.com (B.Y.); 3Ningxia Institute for Science and Technology Development Strategy and Information Research, Yinchuan 750002, China; lanchong.77@163.com

**Keywords:** high-hydrostatic-pressure technology, Marselan, polyphenols, volatile aromas, flavor profile

## Abstract

This study investigated the effects of high-hydrostatic-pressure (HHP) treatment of varying intensity (100–600 MPa) and duration (10–30 min) on polyphenols and volatile aromatic compounds in Marselan red wine. The types and concentrations of polyphenols and volatile aromatic compounds were compared before and after HHP treatment; the results indicated that HHP treatment at 300 MPa for 20 min significantly increased the total polyphenol content to 369.70 mg/L, a rise of 35.82%. The contents of key polyphenols, such as resveratrol and protocatechuic acid, were significantly enhanced. Furthermore, while the total content of volatile aromatic compounds did not change significantly under this condition compared to the untreated samples, the concentration of ester compounds significantly increased to 1.81 times that of the untreated group, thereby enriching the floral and fruity aromas of the wine and effectively improving its aromatic profile and sensory quality. Principal component analysis (PCA) further validated the positive impact of HHP treatment on the flavor characteristics of Marselan red wine. These findings provide technical support for the use of HHP in improving wine quality.

## 1. Introduction

As modern consumers increasingly demand higher quality and health benefits from their wines, the wine industry is exploring new technologies to enhance the flavor and nutritional value of their products. Marselan wine stands out due to its distinctive flavor profile, merging the robust structure of Grenache grapes with the elegance of Cabernet Sauvignon. This wine features intense fruit aromas and a light body, with a soft palate [1].

Polyphenolic compounds are crucial for shaping the quality of Marselan red wine. These compounds fall into two main categories, flavonoids and non-flavonoids, which together determine wine color, taste, aroma, and health benefits [2]. Flavonoid polyphenols, including anthocyanins, flavonols, and flavanols, establish the essential color base and provide antioxidant properties [3]. Anthocyanins ensure that the wine displays a range of colors from light red to deep purple. Flavonols, such as catechins and procyanidins, primarily contribute to the texture and astringency of wine through their tannin properties [4]. Flavanols enhance the antioxidant capacity of wine by providing additional protection [5]. Non-flavonoids include stilbenes, such as resveratrol, and phenolic acids, which encompass hydroxybenzoic and hydroxycinnamic acids. Their effects are multifaceted, influencing the taste, aroma, and health benefits of wine [6]. Phenolic acids, such as gallic acid and caffeic acid, influence the antioxidant activity and color stability of wine and add complexity to its aroma through their unique chemical properties [7]. Tannins are crucial in shaping the taste of Marselan red wine; their astringent qualities not only define the wine texture but also play a significant role in its aging potential [8].

The types and concentrations of volatile aromatic compounds play crucial roles in defining the flavor characteristics of wine. These aroma compounds, including esters, higher alcohols, terpenes, aldehydes, ketones, and sulfur compounds, constitute the rich aroma spectrum of wine [9]. Esters are important for their contribution to the fruity and floral scents of apples, pears, red berries, and roses, which significantly affect the sensory quality of wine [10]. Higher alcohols add complexity to the aroma of wine, with delicate scents of roses, almonds, and ripe fruit [11]. Although terpenes are present in small amounts in grapes, they contribute significantly to the citrus and floral aromas of wine. Aldehydes and ketones impart the unique aromas of almonds, grass, and honey to wine [12].

In the field of food processing, high hydrostatic pressure (HHP) has gained attention owing to its effectiveness in sterilizing food while preserving its nutritional and sensory qualities. This non-thermal method involves the application of pressures of up to 600 MPa, which destroys microbial cell walls without heating the product, thereby extending its shelf life and maintaining or enhancing the nutritional value and flavor of the food [13]. Recent studies have demonstrated significant advancements in the application of HHP for the processing of juices and wines. For instance, Chen et al. [14] found that the treatment of kiwi juice at 500 MPa for 15 min preserved its original fruity, fresh, and green aromas. Similarly, the treatment of orange juice at various pressures and temperatures led to the degradation of limonene, significantly increasing the concentrations of alpha-pinene and myrcene and thus enhancing the flavor of the juice [15]. Sun et al. [16] investigated the impact of pressure and time on the sensory qualities of wine and noted improvements in appearance, aroma, and taste after HHP treatment, with the best sensory quality achieved at 500 MPa for 30 min. However, research on the effects of HHP on polyphenols and volatile aroma compounds in Marselan wine is limited, particularly regarding the potential mechanisms for preserving and enhancing wine flavor.

In this study, different conditions of HHP treatment (both the pressure level and duration) were applied to Marselan red wine to investigate their effects on the polyphenolic and volatile aromatic compounds in wine. The goal was to enhance the flavor quality of Marselan wine through HHP treatment and to provide empirical data and technical support for its application in Marselan-wine production.

## 2. Materials and Methods

### 2.1. Materials and Reagents

Marselan grapes were harvested in October 2022 from the Mengshaquan Winery in the eastern foothills of the Helan Mountains, Ningxia. The grapes had a sugar content of 253.65 g/L (measured as glucose), an acid content of 5.33 g/L (measured as tartaric acid), and a pH of 3.48.

The yeast used for commercial winemaking was Lalivin RC212 from Lallemand, Blagnac, France. Other reagents and materials were procured as follows: octanol (standard sample) from Sigma, Ronkonkoma, NY, USA; pectinase (food grade) from Lallemand, France; sodium chloride (analytical grade) and potassium metabisulfite (food grade) from Aisai Co., Ltd. (Tokyo, Japan); citric acid and phosphoric acid (both analytical grade) from Tianjin Beichen Fangzheng Reagent Factory, Tianjin, China; mixed organic acid standards (malic, lactic, citric, tartaric, succinic acids) from Solabio Biotechnology, Beijing, China; and methanol (chromatography grade) from Dikma Technologies, Beijing, China.

### 2.2. Instrumentation and Equipment

The instrumentation and equipment used were as follows: GC-MS: TQ8050 NX, a gas chromatography–mass spectrometry system from Thermo Fisher Scientific, Waltham, MA, USA; Thermo TSQ ALTIS, an ultra-high-performance liquid chromatography–mass spectrometry system (LC/MS/MS); LC-15C, a high-efficiency liquid chromatograph from Shimadzu Corporation, Kyoto, Japan; L2-600/1, a high-pressure system from Huatiansemiao Biotech Engineering Co., Ltd., Tianjin, China; ST-50H, a constant-temperature shaker from Guansen Biotech, Shanghai, China; and BKQ-B50II, a vertical autoclave from Shandong Bokang Disinfection Equipment Co., Ltd., Zhucheng, China.

### 2.3. Determination Methods

#### 2.3.1. Marselan Red-Wine-Making Process

Grapes were destemmed, crushed, and placed in 10 L glass fermentation tanks, filling them up to 80% of their volume. At this stage, 50 mg/L of potassium metabisulfite and 20 mg/L of pectinase were added to the must, followed by 24 h cold maceration at 8 °C. Fermentation was initiated by introducing 200 mg/L of RC212 yeast until the residual sugar level dropped below 4 g/L. Once alcoholic fermentation was complete, the skin was separated, and malolactic fermentation occurred at room temperature. After fermentation, wine was separated from the lees, and potassium metabisulfite was added to it. The wine was bottled and then immediately treated with high pressure, followed by an analysis of its physicochemical properties and sensory quality, and each parameter was tested three times.

#### 2.3.2. High-Hydrostatic-Pressure Treatment

Subsequently, 100 mL of Marselan red wine was added to an aluminum-foil bag and sealed twice under vacuum. This sealed bag was stored at 4 °C in a constant-temperature chamber. Before beginning the experiment, the bath was preheated. Once the temperature reached the 20 °C set point, the sample was placed in the pressure chamber. The treatment parameters were adjusted by setting the pressure levels from 100 to 600 MPa (in steps of 100 Pa) with treatment durations of 10, 20, and 30 min. Each sample underwent HHP treatment once, using different pressure and time combinations. The samples are labeled using the format ‘L—treatment pressure—treatment time’. The original Marselan wine that had not undergone ultra-high-pressure processing was used as the control group.

#### 2.3.3. Determination of Phenolic Acids

The analysis was conducted using an LC-15C high-performance liquid chromatograph equipped with a diode-array detector and SIL-10 AF autosampler from Shimadzu Corporation, Japan. The chromatographic column used was a Waters xTerra MS C18 reversed-phase column, measuring 250 mm × 4.6 mm with a 5 μm particle size. Mobile phase A consisted of 2% aqueous acetic acid solution by volume, and mobile phase B was a mixture of 0.5% aqueous acetic acid solution and acetonitrile in a 50:50 volume ratio.

The detection wavelengths were set at 280 nm for flavan-3-ols and dihydrochalcones, 320 nm for hydroxycinnamic acids, and 360 nm for flavonols. The qualitative analysis involved a comparison of the retention times of the target compounds with those of the standards, whereas quantification was based on external calibration using peak-area measurements.

#### 2.3.4. Determination of Volatile Aromatic Compounds

Headspace solid-phase microextraction (HS-SPME) was used to identify aromatic compounds in wine. The addition of 1.5 g of NaCl and 5 mL of wine sample to a vial was followed by adding 40 μL of 4-methyl-1-pentanol at a concentration of 394.08 μg/L as an internal standard. The vial was sealed, and a CAR/DVB/PDMS extraction fiber was used for adsorption at 45 °C for 30 min. After adsorption, the fiber was inserted into the injection port of a gas chromatograph (GC) and desorbed at 250 °C for 3 min. Finally, the samples were analyzed using gas chromatography–mass spectrometry (GC-MS).

#### 2.3.5. Characteristic Aroma and Aroma-Profile Analysis

The contribution of different volatile aromatic compounds to the aroma of dry Marselan red wine was assessed using odds activity values (OAVs) [17]. Aroma categories were defined according to the wine aroma wheel, and specific descriptions from this study were integrated to classify the aromas into six types: herbal, fruity, floral, fatty, chemical, and baked. The aroma profile of the Marselan wine was constructed by summing the OAVs of the key aroma compounds within each category.

### 2.4. Sensorial Analysis

After completing various HHP treatments on all wine samples, sensory analyses were conducted. These analyses were performed by a blind tasting panel comprising 15 experts from the Institute of Agricultural Product Quality Standards and Testing Technology. Each wine sample (20 mL) was assigned a randomly generated three-digit code. An 8-point structured numerical scale was utilized to comprehensively assess multiple sensory attributes of the wine samples. Each treatment group underwent three separate sensory evaluations. The evaluation criteria included appearance (color, intensity), aromas (herbal, fruity, floral, fatty, baked, and chemical), and taste (acidity, astringency, aftertaste, global impression).

### 2.5. Data Processing and Analysis

Data were compiled in Excel and analyzed for significance using SPSS 26.0. Graphs were created using the R language, TBtools II, and SIMCA 14.1 software.

## 3. Results

### 3.1. The Impact of High-Hydrostatic-Pressure Treatment on the Polyphenol Content of Marselan Red Wine

#### 3.1.1. The Effects of HHP on the Total Polyphenol Content and Types

Phenolic compounds are essential for shaping the structure and essence of wine and influencing its color, hue, and astringency. These properties make phenolics a critical metric for assessing the quality of wine flavors [18]. Table 1 shows that 41 types of polyphenols were identified in the wine samples. Compared with the control group, the polyphenol content in the pressure-treated samples treated with HHP generally increased, showing an initial increase followed by a decrease. Specifically, the total polyphenol content peaked at 300 MPa after 20 min and was 104.32 mg/L higher than that of the control. The likely cause is the disruption of cellular structures by HHP, which enhances the release of polyphenols from the cells [19]. Additionally, as the pressure increases, the polyphenol content may degrade due to the high-pressure treatment, as seen in the fluctuations observed after 10–20 min at 600 MPa (as also found by Liu et al. [20]). Further analysis was required to understand the specific reasons for this discrepancy. To further explore the impact of HHP on the polyphenol composition and content of Marselan wine, we analyzed the changes in various phenolic substances in different treatment groups, focusing on pressure intensity and duration.

#### 3.1.2. Effects of High-Hydrostatic-Pressure Treatment on Various Phenolic Substances

Polyphenols in wine can be classified into two main categories: flavonoids and non-flavonoids. As listed in Table 1, two flavonoids, nine flavonols, and three flavanols were identified among the flavonoid compounds in the samples. Among the non-flavonoid compounds, ten types of hydroxybenzoic acids, seven types of hydroxycinnamic acids, and one type of stilbene were identified. Notably, flavanols were the most abundant phenolic compounds, comprising 57.28–67.38% of the total phenolic content.

The essence of HHP technology is a basic thermodynamic variable. Its core revolves around the transformation of material and energy within the system due to high pressure during processing, where pressure is the primary variable [16]. Studies (Figure 1) showed that if the duration of pressure application was constant under various pressure conditions, the total flavonoid content increased significantly after HHP treatment (except at 100 MPa); it displayed a pattern of an initial increase, followed by a gradual decrease over time (10, 20, and 30 min). Specifically, when the treatment duration was held constant at 10 and 30 min, the total flavonoid content in the wine samples peaked at a pressure of 400 MPa, recording concentrations of 239.43 mg/L and 249.15 mg/L, respectively. These values represent increases of 44.06 mg/L and 53.79 mg/L over the control group. Conversely, when the treatment duration was set at 20 min, the peak total flavonoid content was observed at 300 MPa, amounting to 286.63 mg/L—an increase of 91.27 mg/L compared to the control. Notably, at 20 min under a pressure of 300 MPa, the content of flavonols such as rutin and quercitrin and flavanols like cianidanol and epicatechin increased by 73.39% and 41.55%, respectively, compared to the control. Additionally, the total anthocyanin content also reached its peak under these conditions, with a 71.34% increase compared to the control (*p* < 0.05). In particular, the concentrations of oenin chloride and cyanidin chloride increased from 7.64 and 0.61 mg/L in the control to 9.29 and 3.75 mg/L, respectively (*p* < 0.05). However, myricetin, quercetin, and kaempferol showed varied changes after different pressure treatments, with notable increases in myricetin and quercetin at 200 MPa, measuring 3.29 mg/L and 11.35 mg/L, respectively. Similarly, kaempferol exhibited significant increases at 100 MPa, with increments of 13.41 mg/L and 8.08 mg/L over the control.

The content of non-flavonoid compounds generally decreased with increasing pressure. However, when the treatment time was 20 min, the total amount of non-flavonoid compounds significantly increased to 99.45 mg/L at 300 MPa, an increase of 13.05 mg/L compared to the control. Moreover, under the same treatment conditions, resveratrol and the content of hydroxycinnamic acid compounds, such as ferulic acid, substantially increased, reaching 2 to 3.5 times that of the control group. Additionally, at a pressure of 100 MPa with a treatment duration of 20 min, the concentration of hydroxybenzoic acid in the wine sample peaked at 72.45 mg/L. Conversely, when the treatment duration was fixed at 10 min and 30 min, the total non-flavonoid content gradually decreased as pressure increased to 600 MPa, reaching lows of 61.34 mg/L and 72.17 mg/L, respectively. This decline was primarily due to a significant reduction in ferulic acid content, which decreased by 50% in the 10 min treatment group, while in the 30 min group, it was markedly lower, almost to trace levels.

As shown in Figure 2, the total quantity of flavonoid compounds increased and then decreased as the treatment time increased when the pressure was set between 100 and 300 MPa, reaching a peak at 20 min. Increasing the pressure to 400 MPa stabilized the compound concentration over time. Specifically, at pressures of 100 and 200 MPa, the concentrations of phenolic compounds, such as protocatechuic acid, apigenin, isorhamnetin, and quercitrin, decreased after high-pressure processing, with only minor changes observed in other compounds. At 300 MPa, only protocatechuic acid and cynaroside decreased after processing, whereas most other compounds initially increased and then decreased. At this pressure, the concentrations of cianidanol, epicatechin, and rutin increased by 51.20 mg/L, 16.83 mg/L, and 9.37 mg/L, respectively, after 20 min compared to the control. At 400 MPa, the concentrations of cianidanol and rutin peaked at 30 min, increasing by 31.40 mg/L and 8.37 mg/L, respectively, compared to the control. However, a further increase in the pressure to 500 MPa resulted in no significant differences between the treatment groups.

#### 3.1.3. PCA Analysis

PCA was performed using the R software (version 4.3.1) to differentiate variations in polyphenols among the different HHP treatment groups. The analysis (Figure 3) showed that the first principal component (PC1) and second principal component (PC2) accounted for 42.20% and 20.04% of the total variance, respectively, totaling 62.24%.

In the PCA model, the positive axes of PC1 and PC2 primarily reflected the presence of compounds such as p-coumaric acid, quercetin, myricetin, and ethylparaben. The negative axis of PC1 and the positive axis of PC2 show information related to 3-hydroxycinnamic acid, coumaric acid, quercitrin, and vanillic acid. The positive axis of PC1 and the negative axis of PC2 were strongly associated with compounds such as epicatechin, caffeic acid, protocatechuic acid, resveratrol, and ferulic acid. Conversely, the negative axes of both PC1 and PC2 reflected less information about polyphenolic substances. Specifically, wine samples L-300-20, L-CK, and L-600-10 were primarily located on the positive axis of PC1 and the negative axis of PC2, with the L-300-20 treatment group showing the highest score on PC2, indicating a strong correlation with epicatechin, protocatechuic acid, resveratrol, and ferulic acid. This suggests that treatment under 300 MPa pressure for 20 min can enhance the content of polyphenols such as resveratrol and ferulic acid. On the positive axes of PC1 and PC2, the treatment groups L-600-20, L-300-10, and L-500-10 had similar scores, indicating weak differentiation and a primary association with p-coumaric acid and quercetin. Treatment groups L-100-20, L-500-30, and L-600-30 were positioned far from the other groups, with L-100-20 scoring high on PC1, implying a significant increase in polyphenols such as 3-hydroxycinnamic acid and coumaric acid.

### 3.2. The Effect of High-Hydrostatic-Pressure Treatment on Volatile Aromatic Compounds

#### 3.2.1. The Influence of High-Hydrostatic-Pressure Treatment on the Variety and Content of Volatile Aromatic Compounds

To further analyze the impact of HHP processing on the flavor of Marselan dry red wine, this study utilized solid-phase microextraction–gas chromatography–mass spectrometry (SPME-GC-MS) to analyze and measure the volatile substances in the wine samples. The analysis (Figure 4a) identified 58 volatile compounds in the untreated Marselan dry red wine, including 18 alcohols, 8 ethyl esters, 1 aldehyde, 6 acids, 3 terpenes, and 3 other compounds. After HHP processing, the variety of volatile flavor components in wine increased significantly, especially after treatment at 400 MPa for 20 min, where the number of volatile components increased to 80. Furthermore, HHP processing significantly affected the content of volatile components. In particular, ester compounds increased from 105.00 mg/L to 123.01 mg/L and showed the most significant change. As the processing pressure and time increased, the total ester-compound content first increased significantly and then decreased slightly. At 300 MPa for 20 min, the ester-compound content reached its peak at 12.3 mg/L, which was 1.81 times higher than that in the untreated samples (Figure 4b).

#### 3.2.2. The Impact of High-Hydrostatic-Pressure Treatment on Major Volatile Aroma Compounds

The OAV serves as a metric for assessing the contribution of individual aroma compounds to the overall aroma of wine; an OAV value greater than 1 indicates that the aroma of the compound can be perceived by humans [18]. This study (Table 2) analyzed 19 sets of wine samples treated with different levels of HHP and found that the concentrations of 17 volatile aroma compounds, primarily esters, phenols, and alcohols, exceeded their sensory activity thresholds (OAV > 1). To further explore the effect of HHP treatment on these volatile compounds, we analyzed compounds with an OAV > 1 relative to the pressure and time dimensions.

As shown in Figure 5, the content of each volatile substance in the three sample groups increased with pressure. Specifically, when the treatment times were 10 min and 20 min, the content of volatile substances reached peak values of 81.99 mg/L and 103.88 mg/L, respectively, when the pressure increased to 300 MPa. However, when the treatment time was extended to 30 min and the pressure was increased to 500 MPa, the volatile compound content reached its highest value (76.02 mg/L). However, upon further increasing the pressure to 600 MPa, the content of volatile compounds in all three sample groups decreased compared to their respective peak values by 9.21%, 46.89%, and 23.23%, respectively. Notably, ester compounds, which are typical byproducts of ethanol fermentation, significantly contribute to the floral and fruity aromas of wine. When the treatment pressure was set to 100 and 200 MPa, the contents of ester compounds, such as isoamyl acetate, ethyl acetate, and ethyl caprylate, in the three sample groups did not change significantly. However, as the pressure increased to 300 MPa, the ester content rapidly increased to its maximum value. However, excessively high pressures could affect the stability of ester compounds, leading to a reduction in their content. The findings of this study echo the results of previous research, such as those reported by Yachao [21] in their wine studies. It has been reported that an increase in pressure causes changes in the volume of the treatment system, which may alter the chemical bond structure within the substance, thereby accelerating the chemical reaction rate of volatile substances and ultimately causing changes in the content of volatile substances. However, when the pressure is increased to a certain threshold, the reverse effect may occur [22].

An appropriate extension of the HHP treatment time also contributed to the release of volatile compounds from Marselan wine (Figure 6). Under different HHP conditions, the ester compounds in the six sample treatment groups increased by 14.28 mg/L to 43.73 mg/L compared to the control group. In particular, at pressures of 100 and 200 MPa, the ester content showed an increasing trend with the extension of the treatment time. However, when the pressure increased to 300 MPa, the ester content initially increased and then decreased (peaking at 20 min). Upon further increasing the pressure to 400 MPa, the change in ester compounds was no longer significant and showed a downward trend. The synthesis of ester compounds consumes a certain amount of chemical energy, and HHP treatment promotes this process by increasing the energy in the wine body [23]. At shorter treatment times, the chemical energy accumulated in the wine may not have been sufficient, possibly not reaching the activation energy required to induce ester-compound synthesis. When the treatment time reached 20 min, the chemical energy accumulated in the wine exceeded the energy required for the synthesis of ester compounds, leading to a significant increase in total ester content. As the treatment time was extended further, the change in the total ester content tended to stabilize. Panosyan and others [24], in their study of different vintages of cognac brandy, also observed that the content of ester compounds gradually increased over time, thereby improving the flavor and taste of the brandy, which aligns with the trend of ester changes in wine in this study. The aldehyde content was significantly higher after treatment durations of 10 and 20 min compared to those observed after 30 min. This observation is likely attributed to the enzymatic reactions responsible for aldehyde formation. HHP treatment has the potential to modify the chemical configuration of enzyme molecules. As the treatment duration increases, this alteration may affect enzyme activity, subsequently influencing the synthesis and release of aldehydes [25].

**Table 2 foods-13-02468-t002:** The impact of different HHP treatments on volatile aromatic compounds with an odor activity value greater than 1.

Compound	Concentration/(μg/L)																			Threshold (μg/L) [26,27,28]
	L-CK	L-100-10	L-100-20	L-100-30	L-200-10	L-200-20	L-200-30	L-300-10	L-300-20	L-300-30	L-400-10	L-400-20	L-400-30	L-500-10	L-500-20	L-500-30	L-600-10	L-600-20	L-600-30	
Phenethyl acetate	411.65 ± 14.26 abc	336.7 ± 18.75 efghi	372.15 ± 3.72 cde	326.17 ± 28.43 fghij	347.21 ± 12.52 defgh	319.7 ± 13.94 ghij	305.46 ± 34.01 ij	367.61 ± 13.25 def	309.6 ± 27.52 hij	291.73 ± 26.26 j	351.83 ± 15.34 defg	362.42 ± 44.54 def	387 ± 23.54 bcd	436.03 ± 19.01 a	420.81 ± 26.28 ab	374.3 ± 46 cde	344.77 ± 22.61 efghi	424.67 ± 25.83 ab	424.62 ± 19.46 ab	250
Isoamyl acetate	6077.36 ± 425.42 defg	5299.79 ± 505.57 gh	5529.27 ± 331.76 efgh	6796.93 ± 475.79 d	9125.4 ± 1075.87 a	5689.13 ± 227.57 efgh	8842.54 ± 755.51 ab	5152.76 ± 472.26 h	8328.87 ± 288.52 abc	8214.77 ± 730.14 bc	5317.67 ± 140.69 fgh	7826.89 ± 135.57 c	6223.01 ± 631.57 de	3912.16 ± 103.51 i	7952.18 ± 993.23 c	8564.63 ± 226.6 abc	6160.22 ± 184.81 defg	6163.35 ± 465.32 def	5350.44 ± 192.91 fgh	30
Ethyl acetate	6251.75 ± 225.41 cd	3100.57 ± 124.02 g	7825.94 ± 414.11 ab	5367.53 ± 568.05 e	3048.15 ± 30.48 g	5672.64 ± 294.76 cde	6133.23 ± 212.46 cd	4115.11 ± 108.88 f	8418.32 ± 252.55 a	5624.09 ± 97.41 de	6199.58 ± 647.26 cd	6242.02 ± 286.05 cd	5659.22 ± 226.37 cde	5809.02 ± 307.38 cde	2249.37 ± 155.84 h	7310.44 ± 633.1 b	7663.53 ± 903.52 b	2918.16 ± 267.45 g	6279.98 ± 439.6 c	7500
Hexyl acetate	594.96 ± 64.35 ij	567.05 ± 5.67 j	1252.47 ± 57.4 d	1806.77 ± 177.95 b	855.65 ± 53.44 fg	592.82 ± 42.75 ij	1028.17 ± 54.41 e	1561.44 ± 133.41 c	1964.5 ± 34.03 a	1040.92 ± 72.86 e	908.89 ± 63.62 fg	795.86 ± 68 gh	697.56 ± 66.54 hi	695.77 ± 31.88 hi	913.86 ± 15.83 f	862.49 ± 22.82 fg	1296.65 ± 44.92 d	642.92 ± 40.15 ij	551.6 ± 9.55 j	1500
Ethyl butyrate	697.9 ± 62.03 efgh	686.52 ± 31.46 fghi	865.64 ± 62.42 b	763.31 ± 7.63 cde	807.26 ± 32.29 bc	566.37 ± 25.95 j	705.96 ± 61.54 defgh	864.11 ± 37.67 b	988.65 ± 69.21 a	675.85 ± 17.88 ghi	717.79 ± 31.29 defg	729.63 ± 71.86 defg	656.83 ± 28.63 ghi	574.91 ± 71.81 j	756.07 ± 27.26 cdef	779.26 ± 54.55 cd	867.7 ± 26.03 b	631.68 ± 31.58 hij	620.78 ± 22.38 ij	20
Ethyl caprylate	4307.13 ± 261.99 g	4117.56 ± 288.23 g	11,211.21 ± 917.68 c	19,550.48 ± 2304.97 b	8025.95 ± 526.3 e	8657.03 ± 687.13 de	9178.65 ± 91.79 de	20,813.62 ± 1456.95 ab	22,059.11 ± 1167.26 a	9547.03 ± 626.04 d	8405.8 ± 303.08 de	6634.99 ± 239.23 f	6398.36 ± 610.36 f	4934.81 ± 85.47 g	6586.9 ± 287.12 f	6395.79 ± 230.6 f	5013.57 ± 279.14 g	2186.4 ± 178.96 h	2011.03 ± 111.97 h	580
Ethyl caprate	974.92 ± 44.68 i	885.3 ± 61.97 i	2895.57 ± 176.13 c	3325.2 ± 185.14 b	1373.03 ± 137.3 fg	872.06 ± 26.16 i	2477.56 ± 178.66 d	3943.38 ± 197.17 a	3159.77 ± 175.93 b	1571.82 ± 149.94 f	2493.99 ± 199.52 d	1056.78 ± 104.08 hi	2339.57 ± 101.98 d	1253.51 ± 57.44 gh	2083.29 ± 157.28 e	1326.56 ± 105.29 g	913.69 ± 48.35 i	604.46 ± 74.28 j	634.6 ± 54.22 j	200
Ethyl hexanoate	4729.15 ± 451.13 fg	3592.88 ± 164.65 i	7394.96 ± 295.8 d	10,261.61 ± 271.5 c	6338 ± 385.53 e	3963.93 ± 79.28 hi	6375.37 ± 292.16 e	12,394.77 ± 1101.67 b	13,665.46 ± 236.69 a	7247.59 ± 452.61 d	5236.19 ± 617.34 f	4569.33 ± 199.17 gh	4512.01 ± 274.46 gh	3533.56 ± 61.2 i	4560.52 ± 373.3 gh	4639.9 ± 202.25 fg	4539.19 ± 433.01 gh	2249 ± 192.15 j	1972.25 ± 161.44 j	14
beta-Damascenone	22.2 ± 1.35 b	22.26 ± 1.68 b	ND	ND	8.5 ± 0.53 e	16.55 ± 1.58 d	ND	ND	ND	ND	ND	15.92 ± 1.27 d	ND	19 ± 0.95 c	ND	19.22 ± 0.77 c	ND	18.57 ± 1.65 c	25.11 ± 0.5 a	0.05
Octanoic acid	1054.48 ± 36.53 g	1780 ± 141.28 e	1015.72 ± 36.62 g	638.34 ± 49.86 jkl	916.87 ± 87.46 gh	848.15 ± 38.87 hi	720.61 ± 12.48 ijk	607.09 ± 37.91 kl	532.93 ± 33.28 l	773.15 ± 73.75 hij	1336.84 ± 114.22 f	1353.03 ± 75.33 f	1363.23 ± 138.35 f	2560.28 ± 67.74 c	2031.7 ± 107.51 d	1466.37 ± 89.2 f	2846.41 ± 113.86 b	4533.71 ± 119.95 a	4604.06 ± 200.69 a	500
Hexanoic acid	1922.98 ± 120.09 de	2603.78 ± 130.19 a	1684.03 ± 44.56 f	ND	1924.32 ± 99.99 de	1658.27 ± 141.68 f	1567.84 ± 68.34 fg	ND	1392.79 ± 127.65 g	1546.81 ± 61.87 fg	1895.46 ± 94.77 e	1969.95 ± 171.74 de	1974.16 ± 175.47 de	2346.08 ± 208.52 b	2119.69 ± 132.37 cd	1996.89 ± 163.45 de	2101.72 ± 105.09 cde	2289.53 ± 160.27 bc	2484.12 ± 155.13 ab	420
Decyl aldehyde	26.77 ± 2.14 e	38.09 ± 2.88 b	31.43 ± 0.54 cd	23.72 ± 0.47 ef	41.36 ± 3.39 ab	23.5 ± 1.92 ef	30.35 ± 2.59 d	25.49 ± 0.92 e	41.29 ± 2.98 ab	21.1 ± 0.56 f	42.24 ± 2.57 a	ND	32.8 ± 3.01 cd	ND	26.08 ± 1.71 e	26.48 ± 2 e	30.52 ± 1.06 d	34.09 ± 2.91 c	ND	10
Nonanal	84.19 ± 6.89 gh	120.79 ± 2.42 cd	84.75 ± 3.69 g	84.93 ± 7.4 g	172.78 ± 12.09 a	93.27 ± 4.27 fg	121.59 ± 9.65 cd	116.8 ± 7.66 cd	125.31 ± 11.14 c	91.84 ± 5.11 g	73.16 ± 5.07 hi	139.39 ± 7.76 b	62.04 ± 2.7 ij	58.9 ± 4.71 j	62.81 ± 2.26 ij	104.41 ± 7.31 ef	144.13 ± 10.09 b	113.08 ± 3.92 de	84.15 ± 6.89 gh	15
Acetaldehyde	ND	229.71 ± 16.08 g	624.23 ± 43.25 b	173.09 ± 5.19 h	427.71 ± 33.41 de	354.24 ± 12.77 f	ND	742.37 ± 63.43 a	699.22 ± 36.33 a	ND	ND	575.65 ± 52.76 c	ND	404.85 ± 38.62 e	432.89 ± 22.91 de	464.87 ± 13.95 d	438.89 ± 19.13 de	ND	227.68 ± 8.21 g	186
Hexanol	2104.3 ± 147.3 c	2757.4 ± 153.53 a	2003.64 ± 173.52 c	1721.97 ± 51.66 d	2589.89 ± 258.99 ab	1899.64 ± 165.61 cd	1868.49 ± 37.37 cd	2077.78 ± 90.57 c	2467.79 ± 24.68 b	1905.93 ± 115.93 cd	1921.48 ± 221.6 cd	2699.52 ± 214.27 ab	2082.46 ± 165.29 c	1972.29 ± 71.11 cd	2103.25 ± 200.64 c	2542.61 ± 110.83 ab	2483.35 ± 99.33 b	2031.43 ± 219.73 c	2099.9 ± 127.73 c	1100
3-Methyl-1-butanol	30,780.32 ± 1231.21 cd	43,358.1 ± 2167.9 a	29,350.09 ± 1344.99 cd	24,963.63 ± 660.48 e	42,270.77 ± 2958.95 ab	28,073.56 ± 0 de	27,938.58 ± 1744.76 de	29,077.82 ± 2195.33 cd	39,643.49 ± 2207.26 b	27,748.16 ± 1209.51 de	28,595.03 ± 2233.34 cde	42,425.17 ± 3818.27 ab	30,888.4 ± 3641.69 cd	29,594.02 ± 1183.76 cd	31,033.24 ± 537.51 cd	40,202.6 ± 1752.39 ab	39,472.74 ± 4121.08 b	30,240.51 ± 2361.86 cd	31,856.12 ± 1592.81 c	3000
Linalool	118.59 ± 11.31 bc	ND	83.37 ± 1.67 ij	117.76 ± 3.53 bcd	112.24 ± 8.77 bcde	93.24 ± 9.46 ghi	54.01 ± 3.74 k	134.67 ± 10.52 a	87.81 ± 6.33 hij	80.46 ± 3.69 j	112.5 ± 6.75 bcde	112.28 ± 6.83 bcde	11.27 ± 1.03 l	105.96 ± 1.84 def	106.15 ± 9.55 cdef	104.72 ± 4.8 efg	121.61 ± 14.34 b	97.64 ± 0.98 fgh	120.24 ± 13.39 b	25

Data are presented as means ± SD (*n* = 3); different letters in the same row indicate significant differences between treatments (*p* < 0.05), ‘ND’ represents no reported odor threshold (the same below).

#### 3.2.3. PCA Analysis

To further compare the effects of different HHP treatments on the flavor of Marselan dry red wine, PCA was conducted on the 17 volatile aromatic compounds detected in the experiment with an OAV > 1. The results are shown in Figure 7, with PC1 contributing 38.93% and PC2 contributing 19.28%, cumulatively accounting for 58.21% of the total.

Samples L-200-10 and L-300-20, located on the positive half-axis of PC1 and the negative half-axis of PC2, mainly reflected ester compounds with herbaceous, floral, and fruity aromas. Among them, L-300-20 showed a high positive correlation with ethyl acetate (fruity aroma), hexyl acetate (pear and other fruit aromas), ethyl butyrate (strawberry and apple aromas), and ethyl caprylate (pineapple and pear-throat aromas) [29], indicating that treatment at 300 MPa for 20 min can significantly enhance the fruity and floral aromas of Marselan dry red wine. Samples L-100-20, L-200-30, and L-300-10 were positively correlated with both axes of PC1 and PC2, mainly displaying information on ester compounds, such as ethyl caprate (fruity aroma, fatty flavor) and ethyl hexanoate (green apple aroma) [30], contributing significantly to the floral and fruity aromas of Marselan dry red wine. Samples L-100-10, L-400-20, and L-500-30 scored higher on the negative half-axes of PC1 and PC2, mainly reflecting terpene compounds such as beta-damascenone (sweet apple) [31] and hexanol, which contribute a grassy aroma to the wine. HHP treatment groups such as L-CK, L-500-10, L-500-20, and L-600-10 were more concentrated, distributed on the negative half-axis of PC1 and the positive half-axis of PC2, mainly showing information on compounds such as phenethyl acetate (floral and fruit aromas) [32], with fewer aromatic compounds in the area, indicating that treatment at 500–600 MPa for 10–30 min has no significant impact on the aroma of wine.

### 3.3. Wine Sensorial Characteristics

After conducting a comprehensive sensory evaluation of 18 experimental wine samples and a control group (Figure 8), it was found that the control group exhibited a notably astringent and harsh mouthfeel, with a marked deficiency in aroma. In contrast, wine samples treated with varying degrees of ultra-high pressure displayed diverse sensory characteristics.

In terms of mouthfeel, when the pressure was kept below 200 MPa for a duration of 10 min, the treated wine samples showed a slight reduction in bitterness and less harshness compared to the control group, indicating a modest improvement overall. This suggests that pressures below 200 MPa for a short duration primarily facilitated reactions among certain chemical components in the wine, with minimal impact on the overall structure and composition of the wine. However, at 300 MPa, the effect of different processing durations on mouthfeel was more pronounced. Specifically, a 20 min duration softened the mouthfeel significantly and reduced harshness, whereas a 30 min duration, while further diminishing bitterness, could lead to over-reaction, resulting in an unbalanced and bitter taste [25]. Increasing pressure does not always correlate with quality enhancement; excessive pressure can disrupt the balance between the structural and compositional elements of the wine [33]. For instance, wine samples processed at 400 MPa for 10 min had a slightly inferior mouthfeel compared to those treated at 300 MPa for 20 min; extending the duration to 30 min at pressures of 500 MPa and 600 MPa resulted in an unbalanced and bitter mouthfeel.

Regarding aroma, the aroma profiles of these treated Marselan wines showed similarities, mainly their herbal, fruity, and floral aromas. The intensity of fruit and floral aromas was the most significant, followed by herbal, fatty, and chemical aromas, whereas the baked aroma was relatively weak. Compared to the control group, the HHP-treated grapes showed a significant improvement in all types of aromas, with the increase in fruit and floral aromas being the most notable, without generating off-flavors. This phenomenon is mainly related to an increase in the content of compounds such as ethyl acetate, hexyl acetate, ethyl butyrate, and ethyl caprylate during the treatment process. Based on these results, we inferred that HHP treatment can add rich fruit and floral aromas to Marselan wine.

## 4. Conclusions

In this study, we systematically explored the effects of HHP treatment, under various conditions, on the polyphenols and volatile aromatic compounds in Marselan red wine. These findings revealed that appropriate HHP treatment significantly affected the polyphenolic composition and aromatic characteristics of Marselan wine, thereby effectively enhancing its sensory quality. Notably, under HHP treatment at 300 MPa for 20 min, there was a significant increase in total polyphenol content, especially in key polyphenols such as resveratrol and protocatechuic acid. However, it is important to note that when the pressure was increased to 600 MPa, the polyphenol content decreased, possibly due to structural damage to the compounds caused by the high pressure. This suggests that moderate HHP treatment can effectively release polyphenols from wines. Furthermore, the effects of HHP were evidenced by the synergistic actions of equilibrium, Le Chatelier’s principle, and microstructural ordering, which significantly impacted biomacromolecules, altering their structure and state and thereby modifying their inherent properties. Additionally, given the suboptimal deactivation of enzymes under HHP, the alterations observed in the phenolic compounds and coloration of the wine were closely linked to enzymatic activity [34].

Additionally, under the same conditions, the total amount and types of volatile aromatic compounds in Marselan wine did not change significantly compared to those in the control group. However, the ester-compound content significantly increased under these treatment conditions, bringing richer and more pronounced floral and fruity aromas to the wine. In contrast, when the pressure was raised to 400 MPa and maintained for 20 min, this increase was mainly in quantity rather than in aromatic characteristics, although the total content of volatile components did indeed significantly increase. This indicates that although higher pressures can increase the content of some volatile components in wine, they do not directly lead to changes in aroma complexity or sensory diversity. Analysis has revealed that HHP alters the composition of volatile compounds by modulating the activity of key enzymes in the aroma synthesis pathways, specifically lipoxygenase (LOX) and hydroperoxide lyase (HPL) [35]. Among various metabolic pathways for aroma, the plant-specific fatty acid metabolism pathway is especially crucial. In this pathway, LOX catalyzes the transformation of precursors such as linoleic and linolenic acids into hydroperoxides, which are subsequently converted into aldehydes, alcohols, and esters through the catalytic actions of HPL, alcohol dehydrogenase (ADH), and acyltransferase (AAT) [36]. Research has indicated that HHP has a biphasic effect on LOX: it activates the enzyme at pressures below 400 MPa while significantly inhibiting its activity above this threshold [37]. Consequently, different HHP conditions have markedly distinct impacts on the volatile components of wine.

In conclusion, the optimal sensory characteristics of Marselan wine were achieved under HHP treatment at 300 MPa for 20 min. This condition effectively enhanced the aromatic profile and improved the taste. These findings further validate the potential of HHP treatment for enhancing the quality and flavor of Marselan wine. Future research will explore the effects of HHP treatment on different types of wines and their aging processes and delve into the underlying mechanisms, aiming to achieve customized quality and flavor diversity in wine.

## Figures and Tables

**Figure 1 foods-13-02468-f001:**
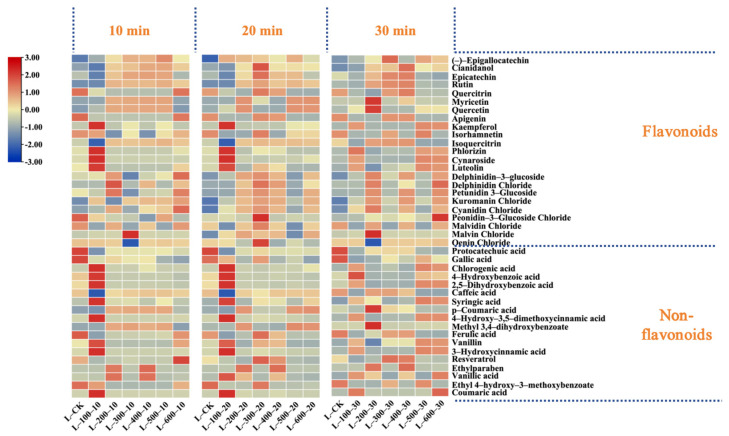
The effect of different HHP treatments on polyphenol content. As the treatment duration remains constant (10 min, 20 min, 30 min), polyphenolic compounds vary with changes in treatment pressure.

**Figure 2 foods-13-02468-f002:**
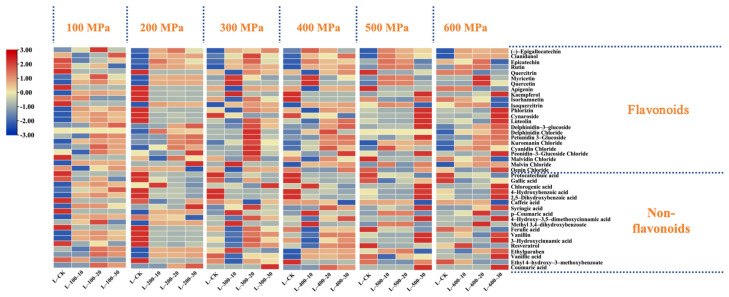
The effects of different HHP treatments on the content of polyphenolic substances. Changes in polyphenolic compounds in wine as the processing time varies with constant pressure (100 MPa–600 MPa).

**Figure 3 foods-13-02468-f003:**
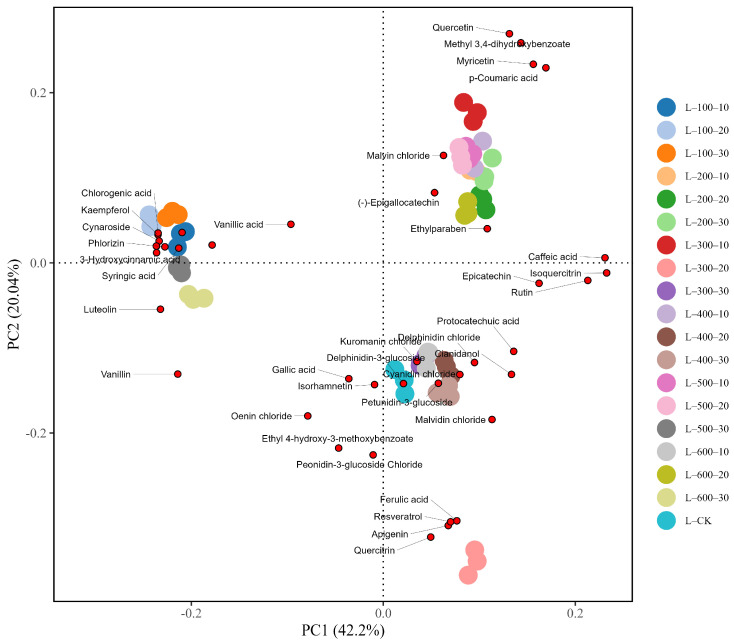
PCA plot of polyphenolic substances under different HHP treatments.

**Figure 4 foods-13-02468-f004:**
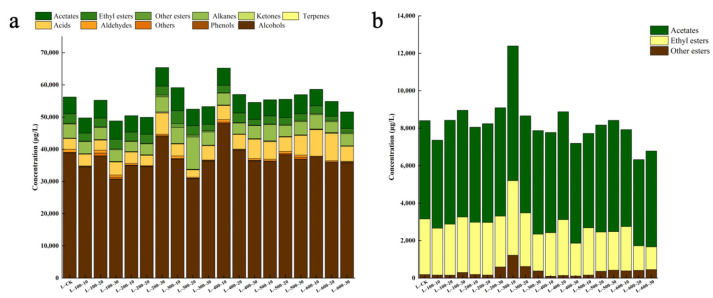
Distribution histogram of aroma substance content in wine samples treated with different HHP processes: (**a**) changes in volatile aroma substances under different HHP treatments; (**b**) changes in ester-type volatile aroma substances under different HHP treatments.

**Figure 5 foods-13-02468-f005:**
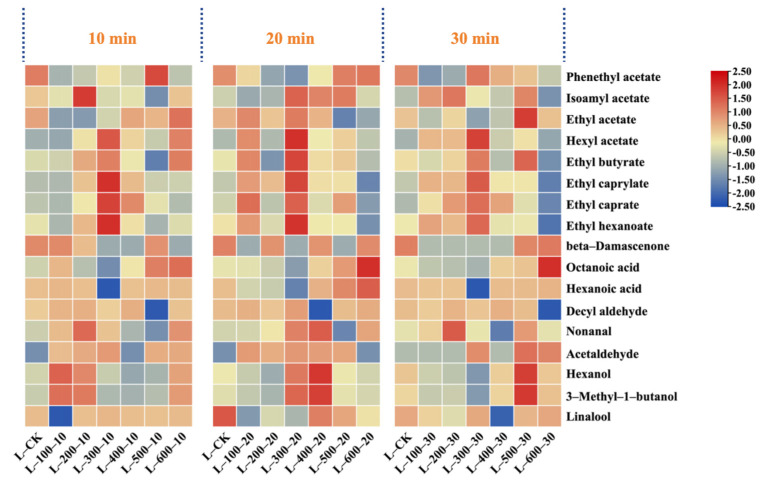
Effects of different HHP treatments on volatile aroma compounds with OAV > 1. As the treatment duration remains constant (10 min, 20 min, 30 min), volatile aroma compounds vary with changes in treatment pressure.

**Figure 6 foods-13-02468-f006:**
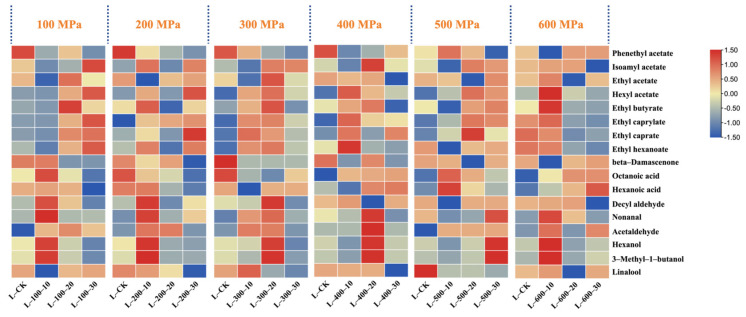
Effects of different HHP treatments on volatile aroma compounds with OAV > 1. Changes in volatile aroma compounds in wine as processing time varies with constant pressure (100 MPa–600 MPa).

**Figure 7 foods-13-02468-f007:**
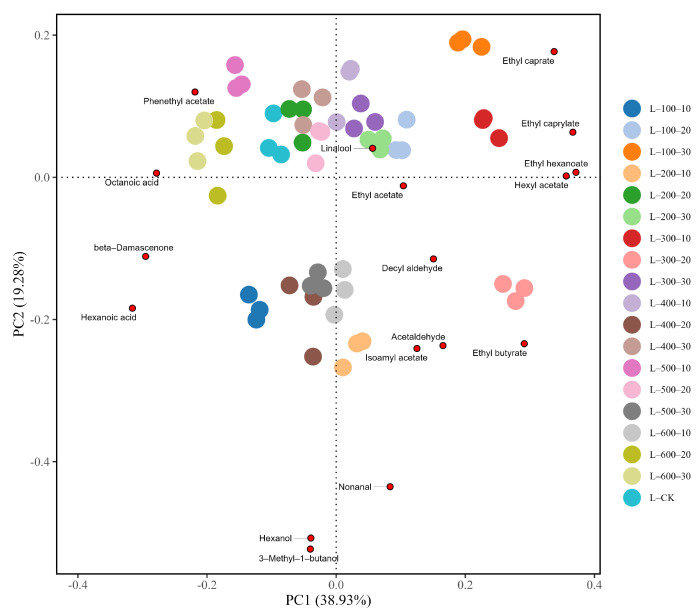
PCA plot of volatile aromatic substances with OAV > 1 under different HHP treatments.

**Figure 8 foods-13-02468-f008:**
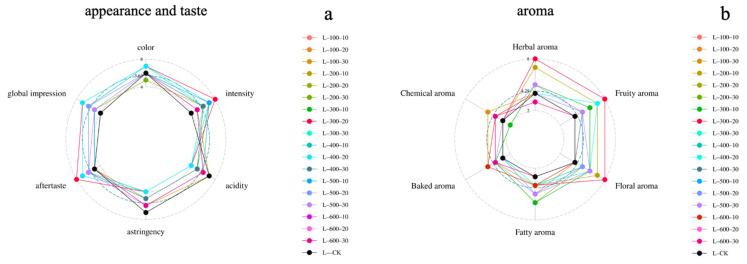
Average sensory scores of wine samples under Different HHP treatments: (**a**) appearance and taste; (**b**) aroma.

**Table 1 foods-13-02468-t001:** The effects of different high-hydrostatic-pressure (HHP) treatments on the content of polyphenolic compounds.

Compound		Concentration/(mg/L)																		
		L-CK	L-100-10	L-100-20	L-100-30	L-200-10	L-200-20	L-200-30	L-300-10	L-300-20	L-300-30	L-400-10	L-400-20	L-400-30	L-500-10	L-500-20	L-500-30	L-600-10	L-600-20	L-600-30
Flavonoids	(-)-Epigallocatechin	0.54 ± 0.03 g	0.6 ± 0.01 efg	0.74 ± 0.05 ab	0.59 ± 0.05 fg	0.7 ± 0.07 bcd	0.73 ± 0.04 abc	0.64 ± 0.03 def	0.74 ± 0.06 abc	0.72 ± 0.05 abc	0.78 ± 0.07 a	0.74 ± 0.05 abc	0.69 ± 0.03 bcd	0.59 ± 0.01 fg	0.78 ± 0.02 a	0.74 ± 0.04 ab	0.71 ± 0.06 abc	0.68 ± 0.03 bcd	0.67 ± 0.02 cde	0.68 ± 0.01 bcd
	Cianidanol	86.36 ± 5.25 d	80.33 ± 6.58 d	82.43 ± 2.18 d	89.26 ± 3.09 d	103.3 ± 1.03 c	104.32 ± 3.76 c	104.77 ± 10.94 c	105.09 ± 2.78 c	137.56 ± 12.61 a	105.62 ± 8.25 c	108.58 ± 2.87 bc	103.88 ± 1.04 c	117.76 ± 7.35 b	109.16 ± 1.09 bc	104.67 ± 4.56 c	100.54 ± 2.66 c	103.24 ± 6.28 c	103.59 ± 8.09 c	103.69 ± 2.07 c
	Epicatechin	77.25 ± 6.73 defg	70.16 ± 4.86 g	72.18 ± 5.91 g	74.15 ± 3.4 fg	85.68 ± 4.28 abcd	84.02 ± 3.03 bcde	81.8 ± 4.98 bcdef	88.63 ± 5.81 ab	94.09 ± 4.89 a	85.1 ± 7.27 abcde	89.23 ± 2.36 ab	86.47 ± 7.69 abcd	85.79 ± 0.86 abcd	88.56 ± 4.69 ab	86.77 ± 9.66 abc	74.18 ± 7.42 fg	76.21 ± 5.95 efg	78.6 ± 2.08 cdefg	73.17 ± 6.98 fg
	Rutin	1.28 ± 0.09 h	1.14 ± 0.1 h	1.2 ± 0.1 h	1.03 ± 0.05 h	8.53 ± 0.56 cde	8.15 ± 0.67 def	7.04 ± 0.58 g	8.27 ± 0.5 cdef	10.65 ± 0.65 a	8.88 ± 1.01 bcd	8.27 ± 0.65 cdef	7.61 ± 0.65 efg	9.65 ± 0.68 b	9.19 ± 0.42 bc	8.85 ± 1.17 bcd	1.38 ± 0.13 h	6.89 ± 0.6 g	7.52 ± 0.49 fg	1.19 ± 0.03 h
	Cynaroside	0.16 ± 0.02 d	4.99 ± 0.17 c	6.08 ± 0.63 b	4.65 ± 0.2 c	0.07 ± 0 d	0.07 ± 0.01 d	0.07 ± 0.01 d	0.06 ± 0.01 d	ND	0.01 ± 0 d	0.07 ± 0 d	0.07 ± 0.01 d	ND	0.07 ± 0 d	0.06 ± 0 d	6.45 ± 0.56 a	0.06 ± 0 d	0.31 ± 0.02 d	5.96 ± 0.36 b
	Quercitrin	3.03 ± 0.32 c	0.47 ± 0.04 d	0.51 ± 0.04 d	0.44 ± 0.01 d	0.14 ± 0.01 e	0.15 ± 0 e	0.15 ± 0.02 e	0.14 ± 0.01 e	4.43 ± 0.15 a	3.1 ± 0.16 c	0.15 ± 0.01 e	2.91 ± 0.1 c	4.09 ± 0.19 b	0.16 ± 0.01 e	0.16 ± 0.01 e	0.54 ± 0.03 d	2.96 ± 0.26 c	0.14 ± 0.01 e	0.51 ± 0.06 d
	Myricetin	ND	0.02 ± 0 e	0.04 ± 0 e	0.02 ± 0 e	3.11 ± 0.25 c	3.3 ± 0.09 abc	3.34 ± 0.26 ab	3.25 ± 0.15 abc	0.73 ± 0.04 d	ND	3.42 ± 0.29 a	ND	0.63 ± 0.02 d	3.4 ± 0.03 a	3.14 ± 0.08 bc	0.04 ± 0 e	ND	3.28 ± 0.15 abc	0.02 ± 0 e
	Luteolin	0.41 ± 0.04 g	3.4 ± 0.3 bc	4.25 ± 0.48 a	3.34 ± 0.22 c	0.05 ± 0 h	0.06 ± 0 h	0.05 ± 0 h	0.05 ± 0 h	1.34 ± 0.1 d	0.79 ± 0.04 f	0.06 ± 0 h	0.83 ± 0.02 ef	1.06 ± 0.1 e	0.06 ± 0 h	0.05 ± 0 h	4.06 ± 0.18 a	0.71 ± 0.04 f	0.05 ± 0 h	3.66 ± 0.29 b
	Quercetin	0.07 ± 0.01 d	1.02 ± 0.09 cd	1.25 ± 0.09 c	0.9 ± 0.01 cd	11.3 ± 0.88 ab	11.35 ± 0.63 ab	11.01 ± 0.83 ab	11.07 ± 1.06 ab	0.07 ± 0 d	0.06 ± 0.01 d	11.67 ± 1.18 a	0.06 ± 0.01 d	0.06 ± 0 d	11.62 ± 0.73 ab	11.06 ± 1.46 ab	1.21 ± 0.04 c	0.05 ± 0 d	10.6 ± 0.7 b	1.15 ± 0.11 c
	Apigenin	12.86 ± 0.68 b	0.06 ± 0 e	0.07 ± 0 e	0.06 ± 0 e	0.01 ± 0 e	0.02 ± 0 e	0.01 ± 0 e	ND	13.87 ± 1.08 a	11.78 ± 1.18 c	0.01 ± 0 e	10.73 ± 0.84 d	11.62 ± 0.58 c	ND	ND	0.07 ± 0.01 e	10.17 ± 0.54 d	ND	0.06 ± 0 e
	Kaempferol	0.03 ± 0 d	12.02 ± 0.21 b	13.44 ± 1.42 a	10.82 ± 0.97 c	ND	ND	ND	0.58 ± 0.06 d	0.04 ± 0 d	0.01 ± 0 d	ND	ND	ND	0.62 ± 0.05 d	0.59 ± 0.06 d	13.87 ± 1.47 a	0.01 ± 0 d	0.63 ± 0.08 d	12.48 ± 0.78 b
	Isorhamnetin	0.7 ± 0.06 b	0.64 ± 0.06 c	ND	ND	0.01 ± 0 f	0.01 ± 0 f	0.01 ± 0 f	0.36 ± 0.02 e	0.78 ± 0.04 a	0.58 ± 0.01 d	0.01 ± 0 f	0.01 ± 0 f	0.01 ± 0 f	0.39 ± 0.02 e	0.37 ± 0.01 e	0.77 ± 0.02 a	0.56 ± 0.01 d	0.37 ± 0.04 e	ND
	Isoquercitrin	2.01 ± 0.27 d	ND	ND	ND	4.36 ± 0.15 bc	4.15 ± 0.37 c	4.12 ± 0.18 c	4.16 ± 0.22 c	4.66 ± 0.4 ab	4.4 ± 0.23 bc	4.42 ± 0.13 bc	4.33 ± 0 bc	4.6 ± 0.12 ab	4.9 ± 0.37 a	4.44 ± 0.29 bc	ND	4.16 ± 0.15 c	4.21 ± 0.19 c	ND
	Phlorizin	1.08 ± 0.09 de	13.46 ± 0.94 a	9.16 ± 0.51 b	14.06 ± 1.15 a	0.62 ± 0.02 e	0.59 ± 0.07 e	0.6 ± 0.06 e	0.62 ± 0.01 e	1.31 ± 0.02 d	1.21 ± 0.07 de	0.61 ± 0.01 e	1.16 ± 0.06 de	1.11 ± 0.12 de	0.63 ± 0.03 e	0.59 ± 0.01 e	8.8 ± 0.81 b	1.13 ± 0.07 de	0.61 ± 0.05 e	8.09 ± 0.32 c
	Delphinidin-3-glucoside	0.27 ± 0.03 fg	0.28 ± 0 fg	0.3 ± 0.04 ef	0.31 ± 0.01 def	0.29 ± 0.02 efg	0.35 ± 0.03 abc	0.38 ± 0.03 a	0.26 ± 0.03 g	0.36 ± 0.02 ab	0.32 ± 0.01 cde	0.27 ± 0.02 fg	0.34 ± 0.02 abcd	0.37 ± 0.01 ab	0.28 ± 0.02 fg	0.28 ± 0.02 fg	0.3 ± 0.03 def	0.3 ± 0.03 efg	0.34 ± 0.03 bcd	0.37 ± 0.03 ab
	Delphinidin Chloride	ND	ND	ND	ND	0.18 ± 0.02 b	0.13 ± 0 cd	0.12 ± 0.01 d	ND	0.22 ± 0.01 a	ND	0.09 ± 0 e	0.17 ± 0.01 b	0.04 ± 0 f	ND	ND	ND	0.09 ± 0.01 e	0.1 ± 0.01 e	0.14 ± 0.02 c
	Petunidin 3-Glucoside	0.34 ± 0.03 de	0.33 ± 0.04 de	0.37 ± 0.01 de	0.36 ± 0.04 de	0.38 ± 0.03 de	0.83 ± 0.05 c	0.93 ± 0.05 a	0.31 ± 0.01 e	0.96 ± 0.02 a	0.39 ± 0.02 d	0.33 ± 0.02 de	0.83 ± 0.1 c	0.91 ± 0.07 ab	0.34 ± 0.02 de	0.34 ± 0.02 de	0.37 ± 0.02 de	0.38 ± 0.02 de	0.84 ± 0.02 bc	0.93 ± 0.06 a
	Kuromanin Chloride	ND	0.11 ± 0 gh	0.12 ± 0.01 efg	0.12 ± 0 efgh	ND	0.24 ± 0.01 d	0.27 ± 0.02 ab	0.1 ± 0 h	0.28 ± 0.02 a	0.12 ± 0 efg	0.12 ± 0.01 fgh	0.26 ± 0.01 bc	0.27 ± 0.02 abc	0.11 ± 0.01 fgh	0.11 ± 0 fgh	0.12 ± 0 ef	0.13 ± 0 e	0.26 ± 0.01 cd	0.27 ± 0.01 abc
	Cyanidin Chloride	0.61 ± 0.06 i	0.82 ± 0.06 hi	2 ± 0.22 d	1.58 ± 0.14 e	1.86 ± 0.16 d	3.09 ± 0.14 b	2.52 ± 0.24 c	0.7 ± 0.03 i	3.76 ± 0.33 a	1.05 ± 0.09 gh	1.37 ± 0.1 ef	3.04 ± 0.2 b	1.97 ± 0.17 d	1.53 ± 0.08 e	1.15 ± 0.09 fg	0.84 ± 0.05 hi	2.59 ± 0.21 c	3 ± 0.14 b	1.93 ± 0.1 d
	Peonidin-3-Glucoside Chloride	0.57 ± 0.01 defgh	0.54 ± 0.02 fgh	0.58 ± 0.07 cdefg	0.57 ± 0.04 defgh	0.53 ± 0.05 gh	0.54 ± 0.02 fgh	0.63 ± 0.04 cd	0.53 ± 0.03 gh	1.19 ± 0.09 a	0.61 ± 0.05 cde	0.52 ± 0.01 h	0.52 ± 0.03 gh	0.64 ± 0.02 c	0.55 ± 0.03 efgh	0.55 ± 0.01 efgh	0.6 ± 0.04 cdef	0.52 ± 0.03 gh	0.54 ± 0.01 fgh	0.86 ± 0.01 b
	Malvidin Chloride	0.12 ± 0.01 bc	ND	ND	ND	0.1 ± 0.01 f	0.13 ± 0.01 b	0.12 ± 0.01 cd	ND	0.19 ± 0.01 a	ND	0.08 ± 0.01 g	0.12 ± 0.01 bc	0.11 ± 0.01 ef	ND	ND	ND	0.12 ± 0 cde	0.12 ± 0.01 bc	0.11 ± 0.01 def
	Malvin Chloride	ND	0.12 ± 0.01 c	0.11 ± 0.01 c	0.11 ± 0.01 c	0.13 ± 0.01 c	0.16 ± 0.01 c	7.67 ± 0.67 a	6.86 ± 0.12 b	0.13 ± 0.01 c	0.11 ± 0.01 c	0.13 ± 0.01 c	0.14 ± 0.01 c	0.15 ± 0.01 c	0.11 ± 0.01 c	ND	0.11 ± 0 c	0.13 ± 0.01 c	0.15 ± 0.02 c	0.15 ± 0.01 c
	Oenin Chloride	7.65 ± 0.67 cd	7.23 ± 0.64 cdef	7.53 ± 1 cde	7.64 ± 0.4 cd	6.83 ± 0.84 def	6.91 ± 0.18 cdef	0.15 ± 0.01 g	0.11 ± 0.01 g	9.29 ± 1.1 a	7.83 ± 0.14 bc	6.64 ± 0.33 ef	6.65 ± 0.35 ef	7.72 ± 0.48 cd	7.15 ± 0.19 cdef	7.14 ± 0.76 cdef	7.51 ± 0.53 cdef	6.6 ± 0.11 f	6.8 ± 0.66 def	8.76 ± 0.46 ab
	total content	195.37 ± 7.04	197.75 ± 9.06	202.36 ± 15.81	210.01 ± 13.12	228.17 ± 26.90	229.32 ± 30.34	226.39 ± 7.84	231.91 ± 22.84	286.63 ± 29.09	232.75 ± 11.64	236.79 ± 16.41	230.82 ± 15.14	249.15 ± 22.82	239.61 ± 24.318	231.09 ± 15.15	222.48 ± 8.89	217.68 ± 15.69	222.73 ± 5.89	224.19 ± 15.69
Non-Flavonoids	Protocatechuic acid	1.9 ± 0.12 a	0.13 ± 0.01 e	0.13 ± 0.01 e	0.1 ± 0.01 e	0.6 ± 0.04 c	0.56 ± 0.01 cd	0.57 ± 0.05 cd	0.58 ± 0.06 c	0.78 ± 0.05 b	0.62 ± 0.05 c	0.6 ± 0.05 c	0.57 ± 0.04 cd	0.63 ± 0.04 c	0.63 ± 0.05 c	0.58 ± 0.03 cd	0.13 ± 0.01 e	0.51 ± 0.01 d	0.56 ± 0.01 cd	0.08 ± 0.01 e
	Gallic acid	59.7 ± 4.14 a	46.58 ± 3.05 efgh	54.41 ± 2.18 bc	44.23 ± 4.68 fgh	49.34 ± 3.73 cdef	47 ± 4.31 efgh	45.32 ± 1.63 efgh	47.6 ± 2.52 efgh	56.43 ± 1.49 ab	49.99 ± 3.6 cde	48.26 ± 3.38 defg	46.91 ± 5.22 efgh	47.41 ± 0.82 efgh	43.58 ± 2.72 gh	48.9 ± 0.85 def	53.32 ± 2.44 bcd	42.4 ± 1.94 h	45.5 ± 2.36 efgh	49.22 ± 4.43 cdef
	Chlorogenic acid	0.02 ± 0 e	0.58 ± 0.05 bc	0.66 ± 0.01 a	0.55 ± 0.05 c	0.03 ± 0 e	0.02 ± 0 e	0.02 ± 0 e	0.02 ± 0 e	ND	0.08 ± 0.01 d	0.08 ± 0.01 d	0.02 ± 0 e	ND	0.02 ± 0 e	0.02 ± 0 e	0.67 ± 0.02 a	0.02 ± 0 e	0.08 ± 0 d	0.59 ± 0.02 b
	4-Hydroxybenzoic acid	0.48 ± 0.02 e	1.32 ± 0.03 c	1.92 ± 0.15 b	3.18 ± 0.3 a	0.17 ± 0.02 f	0.17 ± 0.01 f	0.13 ± 0.01 f	0.17 ± 0.02 f	0.16 ± 0.02 f	0.17 ± 0 f	0.17 ± 0.01 f	0.19 ± 0.01 f	0.14 ± 0.01 f	0.18 ± 0.01 f	0.16 ± 0.01 f	1.25 ± 0.1 c	0.13 ± 0.01 f	0.12 ± 0.01 f	1.09 ± 0.04 d
	2,5-Dihydroxybenzoic acid	0.55 ± 0.01 e	2.41 ± 0.21 c	2.82 ± 0.07 a	2.21 ± 0.12 d	0.19 ± 0.01 f	0.19 ± 0.01 f	0.12 ± 0.01 f	0.19 ± 0.02 f	0.19 ± 0.02 f	0.21 ± 0.02 f	0.2 ± 0.02 f	0.15 ± 0.01 f	0.17 ± 0.02 f	0.18 ± 0.01 f	0.19 ± 0.02 f	2.74 ± 0.32 ab	0.15 ± 0 f	0.16 ± 0.01 f	2.66 ± 0.14 b
	Caffeic acid	1.6 ± 0.06 ab	0.02 ± 0 e	0.03 ± 0 e	0.02 ± 0 e	1.41 ± 0.08 c	1.45 ± 0.16 bc	1.55 ± 0.06 abc	1.53 ± 0.06 abc	1.24 ± 0.13 d	1.53 ± 0.08 abc	1.6 ± 0.19 ab	1.52 ± 0.12 abc	1.1 ± 0.1 d	1.62 ± 0.09 a	1.5 ± 0.15 abc	0.02 ± 0 e	1.5 ± 0.01 abc	1.52 ± 0.13 abc	0.03 ± 0 e
	Syringic acid	4.66 ± 0.34 i	11.05 ± 1.01 a	10.92 ± 0.38 a	8.55 ± 0.6 c	5.18 ± 0.49 fghi	4.79 ± 0.08 hi	7.08 ± 0.31 d	5.29 ± 0.32 fghi	6.31 ± 0.54 de	5.08 ± 0.4 fghi	4.74 ± 0.22 hi	5.84 ± 0.2 ef	5.48 ± 0.52 fgh	5.64 ± 0.41 efg	4.98 ± 0.39 ghi	10.82 ± 0.57 ab	4.74 ± 0.17 hi	6.71 ± 0.66 d	10.13 ± 0.41 b
	p-Coumaric acid	0.17 ± 0 cd	ND	ND	ND	2.96 ± 0.16 a	2.99 ± 0.27 a	3.16 ± 0.2 a	3.04 ± 0.32 a	0.53 ± 0.05 b	0.61 ± 0.06 b	3.1 ± 0.25 a	0.53 ± 0.01 b	0.43 ± 0.04 bc	3.13 ± 0.24 a	3 ± 0.21 a	ND	0.56 ± 0.04 b	3.16 ± 0.31 a	ND
	4-Hydroxy-3,5-dimethoxycinnamic acid	0.12 ± 0 d	4.05 ± 0.23 bc	4.5 ± 0.32 a	3.82 ± 0.35 c	0.09 ± 0 d	0.08 ± 0 d	ND	0.09 ± 0.01 d	0.21 ± 0.01 d	0.16 ± 0.01 d	0.09 ± 0 d	0.14 ± 0.01 d	0.17 ± 0.01 d	0.11 ± 0.01 d	0.09 ± 0 d	4.25 ± 0.41 ab	0.14 ± 0.01 d	ND	4.22 ± 0.32 b
	Methyl 3,4-dihydroxybenzoate	0.16 ± 0.01 f	0.16 ± 0.01 f	0.17 ± 0.02 f	0.15 ± 0.01 f	7.46 ± 0.34 cd	7.09 ± 0.43 de	7.74 ± 0.77 bc	8.14 ± 0.28 ab	0.15 ± 0.01 f	0.08 ± 0 f	6.84 ± 0.34 e	ND	0.09 ± 0.01 f	7.77 ± 0.13 bc	8.27 ± 0.46 a	0.18 ± 0.01 f	ND	7.41 ± 0.26 cd	0.18 ± 0.02 f
	Ferulic acid	14.56 ± 1.29 b	0.12 ± 0 f	0.13 ± 0.01 f	0.12 ± 0.01 f	1.08 ± 0.08 e	1.13 ± 0.06 e	1.27 ± 0.06 e	1.15 ± 0.1 e	28.05 ± 1.84 a	8.15 ± 0.64 cd	1.22 ± 0.06 e	8.09 ± 0.21 cd	8.87 ± 0.41 c	1.35 ± 0.13 e	1.17 ± 0.1 e	0.13 ± 0.01 f	7.32 ± 0.48 d	1.27 ± 0.03 e	0.09 ± 0 f
	Vanillin	0.22 ± 0.01 g	1.16 ± 0.06 c	1.29 ± 0.11 ab	1.03 ± 0 d	0.02 ± 0 h	0.01 ± 0 h	0.02 ± 0 h	0.02 ± 0 h	0.7 ± 0.03 e	0.61 ± 0.04 f	0.03 ± 0 h	0.61 ± 0.01 f	0.6 ± 0.05 f	0.02 ± 0 h	0.02 ± 0 h	1.36 ± 0.05 a	0.59 ± 0.04 f	0.02 ± 0 h	1.28 ± 0.15 b
	3-Hydroxycinnamic acid	0.02 ± 0 d	0.61 ± 0 b	0.67 ± 0.08 a	0.56 ± 0.04 c	ND	ND	ND	ND	0.04 ± 0 d	0.02 ± 0 d	ND	0.02 ± 0 d	0.03 ± 0 d	ND	ND	0.67 ± 0.04 a	0.02 ± 0 d	ND	0.62 ± 0.06 b
	Resveratrol	1.64 ± 0.11 e	0.66 ± 0.02 hi	0.95 ± 0.05 f	0.6 ± 0.01 i	0.78 ± 0.05 fghi	0.71 ± 0.05 ghi	0.81 ± 0.08 fghi	0.86 ± 0.07 fgh	4.15 ± 0.25 a	3.5 ± 0.13 b	0.79 ± 0.06 fghi	2.98 ± 0.28 d	3.2 ± 0.2 c	0.8 ± 0.07 fghi	0.82 ± 0.05 fghi	0.94 ± 0.04 fg	2.92 ± 0.32 d	0.76 ± 0.02 fghi	0.84 ± 0.04 fgh
	Ethylparaben	0.01 ± 0 c	ND	0.02 ± 0 c	0.02 ± 0 c	0.53 ± 0.05 b	0.54 ± 0.03 b	0.56 ± 0.03 ab	0.02 ± 0 c	ND	ND	0.59 ± 0.05 a	0.57 ± 0.05 ab	0.6 ± 0.04 a	0.01 ± 0 c	0.01 ± 0 c	ND	ND	0.02 ± 0 c	0.02 ± 0 c
	Vanillic acid	0.03 ± 0 e	0.02 ± 0 e	0.74 ± 0.04 a	0.58 ± 0.06 c	0.36 ± 0.01 d	0.35 ± 0.04 d	0.38 ± 0.03 d	ND	0.02 ± 0 e	0.02 ± 0 e	0.37 ± 0.04 d	0.35 ± 0.02 d	0.38 ± 0 d	ND	0.02 ± 0 e	0.02 ± 0 e	0.01 ± 0 e	ND	0.68 ± 0.09 b
	Ethyl 4-hydroxy-3-methoxybenzoate	0.55 ± 0.05 a	0.39 ± 0.03 c	0.01 ± 0 e	0.01 ± 0 e	0.01 ± 0 e	ND	ND	0.02 ± 0 e	0.5 ± 0.01 b	0.37 ± 0.03 c	0.02 ± 0 e	ND	0.03 ± 0 e	0.02 ± 0 e	ND	0.48 ± 0.04 b	0.33 ± 0.01 d	0.02 ± 0 e	0.02 ± 0 e
	Coumaric acid	ND	0.02 ± 0 de	0.48 ± 0 a	0.36 ± 0.03 c	ND	0.01 ± 0 de	0.02 ± 0 de	ND	ND	0.02 ± 0 de	ND	0.02 ± 0 de	0.02 ± 0 de	ND	ND	0.03 ± 0 d	ND	ND	0.42 ± 0.05 b
	Total content	86.40 ± 10.79	69.29 ± 7.49	79.86 ± 3.19	66.13 ± 3.49	70.21 ± 3.90	67.11 ± 4.08	68.74 ± 1.82	68.73 ± 5.99	99.45 ± 8.14	71.24 ± 6.79	68.70 ± 3.43	68.49 ± 3.62	69.35 ± 2.50	65.06 ± 4.55	69.72 ± 2.09	77.02 ± 2.77	61.34 ± 3.83	67.29 ± 7.49	72.17 ± 5.45

Data are presented as means ± SD (*n* = 3); different letters in the same row indicate significant differences between treatments (*p* < 0.05), ‘ND’ represents no reported odor threshold, the same below.

## Data Availability

The original contributions presented in the study are included in the article, further inquiries can be directed to the corresponding author.

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
