# Peer review of "Effects of High-Hydrostatic-Pressure Treatment on Polyphenols and Volatile Aromatic Compounds in Marselan Wine"

_foods, 2024, doi:10.3390/foods13152468_

Round 1

Reviewer 1 Report

Comments and Suggestions for Authors

The focus of the study is innovative and interesting. However, the work has serious flaws.

The introduction describes well the background and the different aspects involved in the subject. However, important mistakes are included.

For example, the authors indicate ''Marselan wine stands out for its distinctive flavor profile, typically featuring rich notes of blackberries, raspberries, and vanilla, along with subtle hints of spices and toasted oaks'' (lines 32-34). However, “vanilla” and “toasted oaks” are typical aromas from barrel or wood, and not grape varietal aromas. This is an important conceptual confusion, because the literature cited is "Effect of adding grape seed tannins before barrel aging on the aroma of Cabernet Sauvignon and Marselan dry red wine" (lines 426-427).

No references to the anthocyanin contents of the wines are included. These compounds are essential for the quality of red wines. This omission is very important.

I consider that these flaws are conclusive for not accepting the publication of this text.

In addition, the work has other shortcomings and mistakes.

For example, ''Flavonoid polyphenols include anthocyanins, flavonols, and flavonoid compounds'' (lines 37-38)...''Non-flavonoid polyphenols, particularly phenolic acids and tannins'' (lines 43-44). Tannins are flavonoid compounds. '...'the Principal component analysis (PCA) content of protocatechuic acid decreased by 58.68% at 100 MPa'' (lines 192-193). I do not understand this paragraph...

Finally, the figures are very complex and difficult to understand.

Reviewer 2 Report

Comments and Suggestions for Authors

In my opinion, the work is very interesting and possibly with future for practical applications. It is in general well organized and written. However, the authors must revise some parts before being published. I include some comments for the authors.

- In Line 101, the paragraph and BKQ-B50II, a vertical autoclave from is incomplete. 

- Do the conditions to prepare the wine correspond to those for a real wine process? If not, how were they selected?

- Would be the application of pressure possible before the obtention of wine? Why is the treatment applied after 3 months of aging?

- How was the research design planned? Which is the control group?

- In Table 1, the indication x+s as value and standard deviation must be put. Format for this table must be revised, for example, there are numbers with higher size than others (or difficult to read because of numbers and letters together). Nomenclature for samples must be also explained, although it is easy to understand, but there is no information in the methodology. Cyanidanol vs. Cianidanol in text and tables. 

- Colours in Figure 1 are difficult to interpret, especially because of legend (same colour, different compound). If authors selected compounds considering those with higher concentration, would be this behaviour explained? There are no error bars.

- Authors mentioned that Notably, flavanols were the most abundant phenolic compounds, comprising 57.28% and 67.38% of the total phenolic content. At what do these values correspond?

- Authors indicated Table 2 in Line 178, but it is a mistake. Why did the authors refer that HHP technology is a key thermodynamic parameter? Which thermodynamic parameter?

- Can the authors explain me the paragraph However, the Principal component analysis (PCA) content of protocatechuic acid decreased by 58.68% at 100 MPa that they included in Lines 192-193? 

- Also, the maximun value for epicatechin is not reached at 400 MPa and 30 min as the authors mention. There are higher contents at other conditions, being the highest one 94.09±4.89 mg/L at 300 MPa y 20 min. Moreover, at 10 min, the flavonoid content increased from 174.991±12.12 mg/L to 213.481±13.33, 218.843±11.379, 222.81±11.79 and 224.643±13.47 mg/L, respectively, increasing the pressure. Therefore, the authors must revise the analysis of results showed. 

- Why do the authors think that there was no significant change in the total amount of non-flavonoid compounds at 10 and 30 min

- PCA analysis explained the 75% of total variance. Do the authors consider enough for being significant?

- With respect to aroma, is positive the presence of 1.81 times ester compounds in the treated samples than untreated ones? 

- For me, it is not clear the explanation about the chemical energy accumulated in the wine, especially because of the reduction due to increase of time to 30 min. Can the authors revise it?

- In general, are all tables and figures necessary? In some cases, did they repeat results?

- More than half of references are older (more than five years old).

Comments on the Quality of English Language

Authors must do an English revision for minor mistakes.

Reviewer 3 Report

Comments and Suggestions for Authors

The manuscript sent for review is a very interesting study on the possibility of enriching the volatile compound profile and increasing the polyphenol content of Marselan wine by treating the wine with high hydrostatic pressure. The manuscript is quite correctly prepared, nevertheless it should be improved before possible publication.

1. In the reviewer's opinion, too many variables were used in the study. Limiting the intensity of the high pressure or the time of exposure the wine to them could have been minimised. This would have made it easier to discuss the relationships obtained or to draw clearer conclusions.

2. Please explain what guided the choice of variables - intensity of high pressure and time? Were these values tested in other works? What literature data were referred to in establishing them?

3. Sensory analysis of the selected wines was not carried out in the submitted work, this would have given a complete picture of their taste profile - both in terms of instrumental and sensory evaluation. Please include this aspect in the discussion of the results.

4. Please extend the discussion of the results obtained with a more detailed explanation of the relationships obtained both in terms of polyphenol content and volatile compound profile.

5. Figure 1 is vaguely depicted. The same colours of the fields correspond to the presence of other compounds.

6. Please improve the readability of all figures present in the paper.

7. In the conclusion, please write which high pressure and time parameters used in this study could be recommended for use in practice.

Round 2

Reviewer 1 Report

Comments and Suggestions for Authors

The changes made to the manuscript significantly improved this paper. The changes to the Introduction better support the importance of the work. The results are very well presented and their discussion is well focused. The conclusion is in line with the results obtained and is now significantly better worded.

I believe that the article can be published in this form.

Reviewer 2 Report

Comments and Suggestions for Authors

I think that the manuscript has been improved, and the authors have answered at the majority of my questions. Therefore, in my opinion it can be published.

Reviewer 3 Report

Comments and Suggestions for Authors

The manuscript has been revised in accordance with the reviewer's comments. If it is possible to do so, please further enhance the readability of the images included in the manuscript, e.g. by hanging them.